# Population-based cohort data used to assess trends in early resumption of sexual activity after voluntary medical male circumcision in Rakai, Uganda

**Alex Daama**[1,2]*, **Fred Nalugoda**[1], **Edward Kankaka**[1], **Asani Kasango**[1,2], **Betty Nantume**[2], **Grace Nalwoga Kigozi**[1], **Robert Ssekubugu**[1], **Juliana Namutundu**[2], **Absalom Ssettuba**[1], **Tom Lutalo**[1], **Joseph Kagaayi**[1,2], **Gertrude Nakigozi**[1], **Stella Alamo**[3], **Lisa A. Mills**[3], **Geoffrey Kabuye**[3], **Ron Gray**[1,4], **Maria Wawer**[1,4], **David Serwadda**[1,2], **Nelson Sewankambo**[1,2], **Godfrey Kigozi**[1]

**1** Rakai Health Sciences Program, Kalisizo, Uganda, **2** Makerere University School of Public Health, Kampala, Uganda, **3** Division of Global HIV & TB, US Centers for Disease Control and Prevention, Kampala, Uganda, **4** Johns Hopkins Bloomberg School of Public Health, Baltimore, Maryland, United States of America

* daamaalex6141@gmail.com

**Data Availability Statement:** All data files will be available upon request. Data requests may be sent

## Abstract

### Introduction

Voluntary medical male circumcision (VMMC) reduces the risk of heterosexual acquisition of HIV by 50%–60%. The Uganda Ministry of Health recommends abstinence of sex for 42 days after VMMC to allow complete wound healing. However, some men resume sex early before the recommended period. We estimated trends in prevalence and risk factors of early sex resumption (ESR) among VMMC clients in Rakai, Uganda, from 2013–2020.

### Methods

Data from the Rakai Community Cohort Study (RCCS), a cross-sectional study, were analyzed. Data included consenting males aged 15–49 years in RCCS who self-reported having received VMMC between the period of 2013 to 2020. ESR prevalence and associated risk factors were assessed using modified Poisson regression to estimate adjusted prevalence ratios (aPR).

### Results

Overall, 1,832 participants were included in this study. ESR decreased from 45.1% in 2013 to 14.9% in 2020 (p<0.001). Across the first three surveys, ESR prevalence was consistently higher among the married participants than the never married participants, aPR = 1.83, 95% CI: [1.30,2.57]; aPR = 2.46, 95% CI: [1.50,4.06]; aPR = 2.22, 95% CI: [1.22,4.03]. ESR prevalence was higher among participants who reported to have more than one sexual partner than participants with one partner, aPR = 1.59, 95% CI: [1.16,2.20]. In the fourth survey from 2018–2020, ESR prevalence was significantly higher among participants with primary education than participants with post-primary, aPR = 2.38, 95% CI: [1.31, 4.30].

to the Rakai Health Sciences Program data management office (datarequests@rhsp.org), where data are archived across all the various projects run by the RHSP (original paper forms from the RCCS surveys, as well as the electronic datasets for each survey round).

**Funding:** DS The RCCS study was funded by National Institute of Allergy and Infectious Diseases (R01AI110324, U01AI100031, R01AI110324, R01AI102939), the National Institute of Child Health and Development (RO1HD070769, R01HD050180), the Bill & Melinda Gates Foundation (22006.02), and the NIH Fogarty International Center (5D43TW009578–02). "This project has been supported by the President's Emergency Plan for AIDS Relief (PEPFAR) through the Centers for Disease Control and Prevention under the terms of [NU2GGH002009] the funders did not play any role in the study design, data collection and analysis, decision to publish, or preparation of the manuscript.

**Competing interests:** The authors have declared that no competing interests exist.

However, ESR prevalence was lower among participants aged at least 45 years than participants aged 15–19 years, aPR = 0.0, 95% CI: [1.86e-07, 2.69e-06]. Overall, participants who reported primary school as their highest level of education reported ESR more often than those with post-primary education aPR = 2.38, 95% CI: [1.31, 4.30]. Occupation and known HIV status were not associated with ESR.

## Conclusions

Self-reported ESR after VMMC declined between 2013 and 2020. Targeted efforts for counseling focusing on married men, men who had multiple sex partners, and men with lower levels of education may decrease ESR.

## Introduction

Voluntary medical male circumcision (VMMC) stands as a significant intervention in the global battle against HIV/AIDS, with substantial evidence showcasing its effectiveness in reducing the risk of heterosexual HIV acquisition by 50%–60%in men [1–3]. Originating from recommendations by the World Health Organization (WHO), VMMC was embraced by the Uganda Ministry of Health in 2007 [4]. Essential to the success of VMMC is adherence to a critical recovery period, during which men are advised to abstain from sexual activity for 42 days to ensure complete wound healing [5, 6].

In South Africa, VMMC has been integrated into the national response to HIV/AIDS, underscoring its pivotal role in the National Strategic Plan [7]. Despite its recognized potential as a preventive measure, concerns linger regarding health risks associated with the procedure, particularly regarding the increased vulnerability to HIV transmission during the post-circumcision healing period [8]. Studies, such as those conducted in Zambia and Botswana shed light on the intricate dynamics at play, suggesting that early sex resumption (ESR) post-VMMC could potentially negate the program's intended benefits [9, 10].

While efforts in Uganda, supported by initiatives like the President's Emergency Plan for AIDS Relief (PEPFAR), strive to provide VMMC services free of charge and offer comprehensive pre- and post-circumcision counseling. The challenges of early resumption of sex after voluntary medical male circumcision persist among Ugandan men in settings such as Rakai. The recommended six weeks of abstinence after circumcision is perceived as excessively long by many sexually active men. This duration can lead to discomfort and increased temptation, particularly if partners are present, making it difficult for men to adhere to the guidance during the healing period [11]. In Uganda, approximately 5, 350, 707males aged 10 years and above have received Voluntary Medical Male Circumcision (VMMC) services since the program's inception [12]. Overall coverage since its adoption in 2010, VMMC uptake has increased but remains below the recommended targets. As of 2020, only 57.5% of men aged 15–49 had undergone the procedure, falling short of the WHO target of 80% coverage [13, 14]. Post-operative guidance for men who undergo VMMC typically includes recommendations to avoid sexual activity for a specified period to ensure proper healing for example no vaginal sex for at least six weeks post-surgery and avoiding all sexual activity, including masturbation, during the initial healing phase [15, 16]. Previous research underscores the tendency for circumcised men to engage in sexual activity before the recommended healing period, thereby heightening the risk of HIV acquisition and transmission to their partners. Other risks include wound disruption, bleeding, wound swelling, and pain. Cross-sectional studies indicate that

between 25% to 50% of men undergoing VMMC resume sex before the recommended 42 days of abstinence [9, 17–19]. Research indicates that factors associated with early sex resumption include older age, lower education and unemployment [20], being married [17] and polygamy or multiple sex partners [19]. However, temporal changes in the prevalence of ESR and associated factors are unknown.

This study was implemented through the RCCS. The RCCS is an open population-based cohort which enrolled about 20,000 consenting participants 15–49 years of age in 40 communities across the greater Rakai region. The survey occurs every ~18 months capturing data on socio-demographic characteristics, sexual behaviors in the past 12 months, HIV, sexually transmitted infections (STIs) including HIV prevention and treatment services, self-reported circumcision status and non-communicable diseases [21]. HIV testing was part of the VMMC program and HIV-positive men were eligible for VMMC. The communities include agrarian, peri urban/trading, and fishing communities along the shores of Lake Victoria. HIV prevalence ranges from 5.2% in agrarian communities to 44.3% in fishing communities [22]. Our overall objective was to estimate trends in prevalence of ESR and associated risk factors among men undergoing VMMC in Rakai, Uganda; over an eight -year period (2013–2020).

## Materials and methods

### Study design and setting

We used cross-sectional data from the Rakai Community Cohort Study (RCCS) and the same standardized questions were asked in all four rounds of the survey. The RCCS is comprised of four districts including Kyotera, Rakai, Lyantonde and Masaka. Circumcised participants were asked if they resumed sex and if "Yes" when they resumed sex after circumcision (Have you had sex with any one since you were circumcised? If yes, how soon after circumcision did you resume sex?). All variables were self-reported and circumcision status included only participants who self-reported to have been medically circumcised. Participants who reported being circumcised were asked when they received the service and where they received the service.

### Participants

We included males aged 15–49 years who reported circumcision in any of four successive RCCS survey rounds (June 2013-January 2015), (February 2015- September 2016), October 2016 to May 2018), and (June 2018 to October 2020). Participants were censored for all surveys and we considered participants who reported recent circumcision (< = 3 years) at the time of interview or data collection at each survey visit. Formal written consent was obtained from each participant aged 18 years and above prior to data collection. We also obtained assent from all participants less than 18 years prior to participation. All consenting participants who received VMMC services provided blood samples for HIV testing and were archived for future testing. We conducted a multivariate analysis for each survey round.

**Data analysis.** Frequencies and proportions were used to describe participant characteristics by survey round for categorical variables, and means and medians were estimated for continuous variables. At each round, the prevalence and 95% confidence intervals (CIs) were estimated for recently circumcised participants reporting early sex resumption < 42 days (ESR). Risk factor analyses was conducted and the dependent variable was ESR; and independent variables included age, number of sexual partners in the past 12 months, education, occupation, marital status, and HIV status. Modified Poisson regression was used to estimate unadjusted and adjusted prevalence ratios (aPR) of ESR at each round. The multivariable model included variables with p-value < = 5% identified during bivariate analyses. Analyses were performed using Stata version 14.0 (StataCorp, 2015). We checked for effect modification

including multi-collinearity and found that the mean of VIF of 1.32 hence indicating no relationship between factors.

**Ethical considerations.** The RCCS is approved by the Uganda Virus Research Institute (UVRI) Research and Ethics Committee (REC), the Johns Hopkins University School of Medicine (IRB), and the Uganda National Council Science & Technology. All data were de-identified and participants aged 18 years and above had provided written informed consent. Assent was obtained from all participants less than 18 years prior to participation.

## Results

### Changes in socio-demographic characteristic of circumcised participants over four successive survey periods (2013–2020)

From Table 1 below, majority of the respondents were aged (25–34) years representing 37.11% (167/450) in 2013/2015, this was similar across two survey rounds (2016/2018& 2018/2020) reporting 34.19% (212/620) and 27.97% (73/261) respectively, however, this was dissimilar in 2015/2016 representing 29.94% (150/501). This was followed by participants aged (20–24) years in 2013/2015 reporting 28.00% (126/450), this was similar across one survey round (2016/2018) representing 25.81% (160/620). However, this was dissimilar across two survey rounds (2015/2016 & 2018/2020) representing 31.14% (156/501) & 25.67% (67/261)

**Table 1. Shows baseline socio-demographic characteristics over time (2013–2020).**

| Variable | Survey round 2013/2015 | | Survey round 2015/2016 | | Survey round 2016/2018 | | Survey round 2018/2020 | |
|---|---|---|---|---|---|---|---|---|
| | Frequency | %tage | Frequency | %tage | Frequency | %tage | Frequency | %tage |
| **Age group** | | | | | | | | |
| 15–19 | 60 | 13.30 | 92 | 18.36 | 95 | 15.32 | 71 | 27.20 |
| 20–24 | 126 | 28.00 | 156 | 31.14 | 160 | 25.81 | 67 | 25.67 |
| 25–34 | 167 | 37.10 | 150 | 29.94 | 212 | 34.19 | 73 | 27.97 |
| 35–44 | 75 | 16.70 | 83 | 16.57 | 127 | 20.48 | 43 | 16.48 |
| 45+ | 22 | 04.89 | 20 | 3.99 | 26 | 4.19 | 7 | 2.68 |
| **Education** | | | | | | | | |
| Post-primary | 129 | 28.70 | 156 | 31.14 | 167 | 26.94 | 133 | 50.96 |
| Primary | 298 | 66.20 | 334 | 66.67 | 429 | 69.19 | 123 | 47.13 |
| None | 23 | 5.11 | 11 | 2.20 | 24 | 3.87 | 5 | 1.92 |
| **Marital status** | | | | | | | | |
| Never married | 158 | 35.10 | 226 | 45.11 | 238 | 38.39 | 133 | 50.96 |
| Currently married | 236 | 52.40 | 233 | 46.51 | 305 | 49.19 | 94 | 36.02 |
| Not married | 56 | 12.40 | 42 | 8.38 | 77 | 12.42 | 34 | 13.03 |
| **Occupation** | | | | | | | | |
| Trading | 73 | 16.20 | 93 | 18.56 | 82 | 13.23 | 53 | 20.31 |
| Agrarian | 245 | 54.40 | 272 | 54.29 | 319 | 51.45 | 155 | 59.39 |
| Fishing | 132 | 29.30 | 136 | 27.15 | 219 | 35.32 | 53 | 20.31 |
| **HIV status** | | | | | | | | |
| Negative | 384 | 85.30 | 437 | 87.23 | 509 | 82.10 | 228 | 87.36 |
| Positive | 66 | 14.70 | 64 | 12.77 | 111 | 17.90 | 33 | 12.64 |
| **Sexual partners** | | | | | | | | |
| One partner | 220 | 48.90 | 263 | 52.50 | 313 | 50.48 | 117 | 44.83 |
| More than 1 partner | 187 | 41.60 | 226 | 45.11 | 284 | 45.81 | 130 | 49.81 |
| None | 43 | 09.56 | 12 | 02.40 | 23 | 03.71 | 14 | 05.36 |

respectively. This was followed by respondents aged (35–44) years under 2013/2015 representing 16.70% (75/450) and this was only similar with one survey round (2016/2018) representing 20.48% (127/620), however, this was dissimilar across two survey rounds (2015/2016 & 2018) reporting 16.57% (83/501) and 16.48% (43/261) respectively. The fourth age group in 2013/2015 was 15–19 years representing 13.30% (60/450), this finding was only similar with only survey round (2016/2018) representing 15.32% (95/620), however this was dissimilar across two survey rounds (2015/2016 & 2018/2020) reporting 18.36% (92/501) & 27.20% (71/261) respectively. The least age group in 2013/2013 were those aged at least 45 years representing 04.89% (22/450) and this was similar across all the three survey rounds (2015/2016, 2016/2018 & 2018/2020) representing; 03.99% (20/501), 04.19% (26/620) and 02.68% (07/261) respectively.

In terms of education status in 2013/2015, majority of the participants had primary level of education representing 66.2% (298/450) and this was similar in 2015/2016 & 2016/2018 representing 66.67% (334/501) and 69.19 (429/620) respectively, however this was dissimilar in 2018/2020 with 47.13% (123/261). Participants with post-primary level of education in 2013/2015 were the second representing 28.7% (129/450), this finding was similar across two survey rounds (2015/2016 & 2016/2018) reporting 31.14% (156/501) and 26.94% (167/620) respectively. However, this was dissimilar in 2018/2020 which had 50.96% (133/261). Participants without formal education were the least category across all survey rounds ranging from 1.9% to 5.1%.

Regarding marital status, majority of participants were currently married in three rounds except in 2018/2020 representing 52.40% (236/450), 46.51% (233/501) and 49.19% (305/620) respectively. This was only dissimilar in 2018/2020 which had 36.02% (94/261). Participants who never married were the second across three surveys representing 35.10% (158/450), 45.11% (226/501) and 38.39% (238/620) respectively. However, in 2018/2020 this was different representing 50.96% (133/261). All participants who had never married were the least category across all survey rounds representing 12.40% (56/450), 08.38% (42/501), 12.42% (77/620) and 13.03% (34/261) respectively.

In terms of occupation, out of 1,832 circumcised participants, majority were doing agriculture across all survey rounds representing 54.40% (245/450), 54.29% (272/501), 51.45% (319/620) and 59.39% (155/261) respectively. This was followed by participants who doing fishing in all survey rounds representing 29.30% (132/450), 27.15% (136/501), 35.32% (219/620) and 20.31 (53/261) respectively. Traders were the least category, and this was similar across all survey rounds representing 16.20% (73/450), 18.56% (93/501), 13.23% (82/620) and 20.31% (53/261) respectively.

Regarding HIV status, majority of the participants self-reported being HIV negative across all four survey rounds representing 85.30% (384/450), 87.23% (437/501), 82.10% (509/620) and 87.36% (228/261) respectively, while participants with HIV negative results were the least, this was similar across all survey periods representing 14.70% (66/450), 12.77% (64/501), 17.90% (111/620), & 12.64% (33/261) respectively.

Lastly, in terms of number of sexual partners, circumcised participants who self-reported having one partner were the majority across all three survey periods representing 48.90% (220/450), 52.50% (263/501), and 50.48% (313/111) respectively except in 2018/2020 with 44.83% (117/261). Participants who had more than one sexual partner were the second, and this was similar across three survey periods representing 41.60% (187/450), 45.11% (226/501), and 45.81% (284/620) respectively. However, this was dissimilar in 2018/2020 reporting 49.81 (130/261). Finally, circumcised participants who had no sexual partner were similar across all four survey periods representing 09.56% (43/450), 02.40% (12/501), 03.71% (23/620) and 05.36% (14/261) respectively.

**Table 2. ESR prevalence and socio-demographic characteristics over time, 2013–2020, Rakai, Uganda.**

| Socio-demographic characteristics | 2013–2015 | 2015–2016 | 2016–2018 | 2018–2020 |
|---|---|---|---|---|
| | This includes the number that resumed sex<42 days n = 203, Total number resumed N = 450 n/N (%) | This includes the number that resumed sex <42 days n = 109, Total number resumed N = 501 n/N (%) | This includes the number that resumed sex <42 days n = 135, Total number resumed N = 620 n/N (%) | This includes the number that resumed sex <42 days n = 39 Total number resumed N = 261 n/N (%) |
| Overall prevalence of ESR | 203/450(45.11) | 109/501(21.76) | 135/620(21.77) | 39/261(14.94) |
| Age in years | | | | |
| 15–19 | 16/60(26.67) | 9/92(9.78) | 7/95(7.37) | 6/71(8.45) |
| 20–24 | 49/126(38.89) | 29/156(18.59) | 22/160(13.75) | 11/67(16.42) |
| 25–34 | 79/167(47.31) | 42/150(28.00) | 64/212(30.19) | 14/73(19.18) |
| 35–44 | 48/75(64.00) | 25/83(30.12) | 33/127(25.98) | 11/43(25.58) |
| 45+ | 11/22(50.00) | 4/20(20.00) | 5/26(19.23) | 0/7(0.00) |
| Marital Status | | | | |
| Currently married | 135/236(57.20) | 76/233(32.62) | 88/305(28.85) | 22/94(23.40) |
| Previously married | 23/56(41.07) | 7/42(16.67) | 22/77(28.57) | 5/34(14.71) |
| Never married | 45/158(28.48) | 26/226(11.50) | 21/238(8.82) | 15/133(11.28) |
| HIV Status | | | | |
| Negative | 174/384(45.31) | 89/437(20.37) | 95/509(18.66) | 38/228(16.67) |
| Positive | 29/66(43.94) | 20/64(31.25) | 36/111(32.43) | 4/33(12.12) |
| Number of Sexual Partners (in the past 12 months) | | | | |
| One partner in the past 12 months | 89/220(40.45) | 51/263(19.39) | 50/313(15.97) | 16/117(13.68) |
| More than one in the past 12 months | 95/187(50.80) | 58/226(25.66) | 80/284(28.17) | 26/130(20.00) |
| Occupation | | | | |
| Trading | 28/73(38.36) | 18/93(19.35) | 14/82(17.07) | 11/53(20.75) |
| Agrarian | 107/245(43.67) | 59/272(21.69) | 54/319(16.93) | 22/155(14.19) |
| Fishing | 68/132(51.52) | 32/136(23.53) | 63/219(28.77) | 9/53(16.98) |
| Highest Education | | | | |
| Post primary | 49/129(37.98) | 30/156(19.23) | 27/167(16.17) | 13/133(9.77) |
| Primary | 142/298(47.65) | 78/334(23.35) | 95/429(22.14) | 28/123(22.76) |
| No formal education | 12/23(52.17) | 1/11(9.09) | 9/24(37.50) | 1/5(20.00) |

## Characteristics of participants by ESR and survey round

Table 2 presents data on participants who were previously circumcised prior to participation in the survey. The prevalence of ESR declined from 45.1% in 2013/2015 to 21.8% in 2015/2016, and 21.8% in 2016/2018 to 14.9% in 2018/2020 [Fig 1]. Prevalence of ESR was higher among participants aged 35–44 years across three survey rounds; 64.0% in 2013/15, 30.1% in 2015/16 and 25.6% in 2018/20 than other age groups (Table 2).

ESR prevalence was highest among currently married participants across the four surveys, compared to participants who were divorced or separated (Table 2). ESR was more common among participants reporting more than one sex partner than participants with one partner. ESR was higher among fishermen compared to all other occupations across the first three survey rounds in 2013/15, 2015/16, and 2016/18. However, in 2018/20 ESR was lower among fishermen than participants who reported to be engaged in trading.

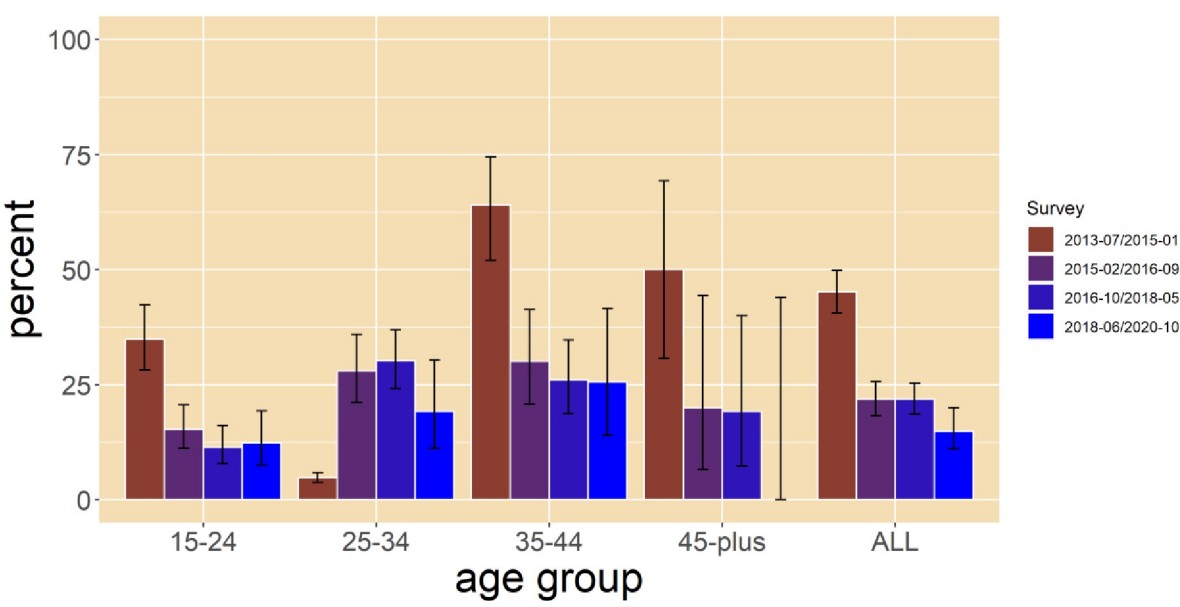

**Fig 1. Trends in ESR per survey round by age group.**

ESR prevalence was also higher among those with primary education in two survey rounds, 23.4% in 2015/16 and 22.8% in 2018/20 compared to those who completed post-primary education and men without formal education.

**Trends in prevalence of ESR stratified by age across survey periods.** We observed similar trends among individuals after stratification by age where ESR declined over time. However, ESR increased from 4.7% (CI = 3.8–5.9) in (2013/15) to 28.0% (CI = 21.1–36.0) in (2015/16) among individuals aged 25–34 years, with a slight reduction to 23.0% in 2018/2020 (Fig 1).

In bivariate analysis, ESR prevalence was similarly higher among those aged 35–44 years than men aged 15–19 years across all the four study periods in 2013/15, 2015/16, 2016/18 and 2018/20 [Table 3]. In addition, ESR prevalence was significantly higher among currently married participants than participants who never married across all surveys (Table 3).

ESR prevalence was significantly higher among HIV-positive participants compared to HIV-negative participants in 2015/2016 and 2016/2018; PR = 1.53, 95% CI: [1.02,2.31]; and PR = 1.74, 95% CI: [1.26,2.40] respectively. However there we no significant difference in rounds 2013/2015 and 2018/2020.

ESR was significant in participants with multiple sex partners than those reporting one sexual partner in one survey round, 2016/2018 aPR = 1.95, 95% CI: [1.48, 2.58]. However, this was not significant in three survey rounds, 2013/15, 2015/16 and 2018/20, aPR = 1.26, 95% CI: [0.94, 1.68], aPR = 1.32, 95% CI: [0.95, 1.84] and aPR = 1.46, 95% CI: [0.83,2.59], respectively. In 2016/2018, ESR was significantly higher among fisher men compared to traders; PR = 1.68, 95% CI: [1.00,2.84] and in 2016/18 ESR was higher among participants with no formal education compared to participants with post-primary education; PR = 2.32 95% CI: [1.25,4.32]. ESR was more prevalent among participants with primary education than participants with post-primary education; PR = 2.33, 95% CI: [1.26,4.29] in 2018/2020.

**Multivariate analysis of risk factors associated with ESR across all study periods (2013–2020).** Table 4 shows the multivariate analysis, in 2018/2020 and after adjusting for marital status, age, HIV status, occupation, and number of sexual partners in the past 12 months ESR

**Table 3. Unadjusted early sexual resumption prevalence ratios post-circumcision by survey round 2013–2020 in Rakai, Uganda.**

| Factor | 2013–2015 | 2015–2016 | 2016–2018 | 2018–2020 |
|---|---|---|---|---|
| | Unadjusted PR[95% CI] | Unadjusted PR [95% CI] | Unadjusted PR [95% CI] | Unadjusted PR [95% CI] |
| Education level | | | | |
| Post-primary | 1 | 1 | 1 | 1 |
| Primary | 1.25[0.98, 1.61] | 1.21[0.83,1.77] | 1.37[0.93,2.02] | 2.33[1.26,4.29] ** |
| No formal education | 1.37 [0.88, 2.15] | 0.47 [0.07,3.15] | 2.32 [1.25,4.32] * | 2.05[0.33, 12.77] |
| Age group | | | | |
| 15–19 | 1 | 1 | 1 | 1 |
| 20–24 | 1.46[0.91, 2.34] | 1.90 [0.94,3.84] | 1.87[0.83,4.20] | 1.94[0.76,4.97] |
| 25–34 | 1.77[1.13, 2.78] | 2.86 [1.46,5.60] * | 4.10[1.95,8.61] *** | 2.27[0.92,5.58] |
| 35–44 | 2.40[1.53, 3.78] *** | 3.08[1.53,6.22] * | 3.53[1.63,7.63] ** | 3.03[1.20,7.67] * |
| 45+ | 1.88[1.04, 3.39] * | 2.04[0.70,5.99] | 2.61[0.90,7.56] | 0[9.50e-07, 8.03e-06] *** |
| Occupation | | | | |
| Trading | 1 | 1 | 1 | 1 |
| Agrarian | 1.14 [0.82, 1.57] | 1.12[0.70,1.80] | 0.99[0.58,1.69] | 0.68[0.36,1.32] |
| Fishing | 1.34 [0.96, 1.88] | 1.22[0.73,2.03] | 1.68[1.00,2.84] * | 0.82[0.37,1.81] |
| Sexual partners in the past 12 months | | | | |
| One partner in the past 12 months | 1 | 1 | 1 | 1 |
| More than 1 partner in the past 12 months | 1.26[0.94, 1.68] | 1.32[0.91, 1.93] | 1.95 [1.48, 2.589] *** | 1.58 [.97, 2.58] |
| HIV status | | | | |
| Negative | 1 | 1 | 1 | 1 |
| Positive | 0.97[0.72,1.30] | 1.53[1.02,2.31] * | 1.74[1.26,2.40] ** | 0.73[0.28,1.91] |
| Marital status | | | | |
| Never married | 1 | 1 | 1 | 1 |
| Currently married | 2.01 [1.53, 2.63] *** | 2.84[1.89,4.26] *** | 3.27[2.10,5.10] *** | 2.08[1.14,3.79] * |
| Previously married | 1.44[0.97, 2.15] | 1.45[0.67,3.12] | 3.24[1.89,5.56] *** | 1.30[0.51,3.34] |

***p<0.001,

**p<0.01 &

*p<0.05

was significantly higher among participants who reported primary education as their highest level of completed education than participants with post-primary education; aPR = 2.38, 95% CI: [1.31, 4.30]. ESR was also significantly higher among participants who were currently married than participants who never married, and this was similar across three study periods in 2013/15, 2015/16, 2016/18 and it was not significant in 2018/2020. In 2016/2018, ESR was also significantly higher among unmarried participants than participants who never married; aPR = 1.96, 95% CI: [0.99,3.90]. In 2016/2018, ESR was higher among participants who had more than one sexual partner than those who had one sexual partner, aPR = 1.76, 95% CI: [1.32,2.34]. However, this finding was not significant across three study survey periods.

ESR was significantly lower among participants ≥45 years than participants aged 15–19 years in 2018/2020, aPR = 0, 95% CI: [9.50e-07, 8.03e-06]. Finally, occupation and HIV status were not associated with ESR across all study survey rounds.

Overall, ESR declined over time (Table 5) from 45.1% in 2013/2015 to 21.8% in 2015/2016; 21.1% in 2016/2018 and 14.9% in 2018/2020.

In addition, recent survey visits in 2016/18 and 2018/20, ESR significantly increased, for instance ESR prevalence was significantly higher among participants aged 25–44 years than participants aged 15–19 years. ESR prevalence was significantly higher among married

**Table 4. Adjusted early sexual resumption post-circumcision by round 2013–2020 in Rakai, Uganda.**

| Factor | 2013/2015 | 2015/2016 | 2016/2018 | 2018/2020 |
|---|---|---|---|---|
| | Adjusted PR (aPR)[95% CI] | Adjusted PR (aPR) [95% CI] | Adjusted PR (aPR) [95% CI] | Adjusted PR (aPR) [95% CI] |
| Education level | | | | |
| Post-primary | 1 | 1 | 1 | 1 |
| Primary | 1.23[0.97,1.56] | 1.17 [0.80,1.71] | 1.12[0.75,1.68] | 2.38[1.31, 4.30] ** |
| No formal education | 1.25[0.80,1.94] | 0.53[0.08,3.61] | 1.53 [0.85,2.75] | 2.21[0.34, 14.50] |
| Age group | | | | |
| 15–19 | 1 | 1 | 1 | 1 |
| 20–24 | 1.22 [0.74,2.00] | 1.36[0.65,2.86] | 1.19[0.50, 2.83] | 1.56[0.64, 3.84] |
| 25–34 | 1.16 [0.70,1.94] | 1.37[0.63,2.97] | 1.79[0.70,4.54] | 1.68[0.62, 4.51] |
| 35–44 | 1.55[0.92,2.63] | 1.38[0.60,3.16] | 1.39[0.53,3.64] | 2.17[0.76, 6.15] |
| 45+ | 1.16[0.59,2.25] | 0.95[0.29,3.11] | 0.99[0.29,3.39] | 7.08e-07[1.86e-07 2.69e-06] *** |
| Occupation | | | | |
| Trading center | 1 | 1 | 1 | 1 |
| Agrarian | 1.05 [0.77,1.44] | 1.25[0.80,1.95] | 1.16[0.69,1.95] | 0.78[0.37, 1.63] |
| Fishing | 1.20[0.86,1.67] | 0.94[0.56,1.58] | 1.30[0.77,2.18] | 0.67[0.28, 1.62] |
| Sexual partners in the past 12 months | | | | |
| One partner in the past 12 months | 1 | 1 | 1 | 1 |
| More than one partner in the past 12 months | 1.18 [0.88,1.59] | 1.26[0.86, 1.85] | 1.76[1.32,2.34] *** | 1.17 [0.70, 1.96] |
| HIV status | | | | |
| Negative | 1 | 1 | 1 | 1 |
| Positive | 0.77[0.57,1.05] | 1.24[0.80,1.92] | 1.08 [0.78,1.50] | 0.57 [0.22, 1.47] |
| Marital status | | | | |
| Never married | 1 | 1 | 1 | 1 |
| Currently married | 1.83[1.30,2.57] ** | 2.46[1.50,4.06] *** | 2.22[1.22,4.03] ** | 1.32[0.61, 2.84] |
| Previously married | 1.29[0.82,2.01] | 1.34[0.59,3.04] | 1.96[0.99,3.90] * | 0.83[0.30, 2.27] |

***p<0.001,

**p<0.01 &

*p<0.05

participants and previously married participants compared to participants who never married. ESR prevalence was significantly higher among participants with primary level of education than post-primary aPR = 2.38, 95% CI: [1.31, 4.30] in 2018/20. ESR prevalence was significantly higher among participants with multiple sexual partners than participants without sexual partners aPR = 1.36, 95% CI: [1.15, 1.61].

## Discussion

We observed twofold decline in ESR following VMMC over the 8-year period (2013–2020). A prevalence greater than 35% reported between 2013–2015 is consistent with the prevalence reported in Kenya between 2008 and 2010 [20] and a relative decline in prevalence noted in Uganda between 2015–2016 and 2016–2018 is comparable with an ESR prevalence reported in Kenya [18]. This decline over time could be due to comprehensive HIV programs which emphasized delayed resumption of sex after VMMC that started in 2016 where counseling services were strengthened [23]. Despite the decline, ESR prevalence remains higher than the

**Table 5. Overall merged ESR prevalence, unadjusted and adjusted early sexual resumption prevalence ratios post-circumcision all four surveys (2013–2020) in Rakai, Uganda.**

| Overall ESR Prev. | Prevalence of ESR | Univariate analysis | Multivariate analysis |
|---|---|---|---|
| | n/N % | PR(95% CI) pvalue | PR (95% CI) pvalue |
| Survey Round | | | |
| 2013/2015 | 203/450(45.11) | 1.0 | 1.0 |
| 2015/2016 | 109/501(21.76) | 0.48[0.40, 0.59]*** | 0.51[0.42, 0.62]*** |
| 2016/2018 | 135/620(21.77) | 0.47[0.39, 0.56]*** | 0.47[0.39, 0.56]*** |
| 2018/2020 | 39/261(14.94) | 0.36[0.27, 0.48]*** | 0.42[0.31, 0.56]*** |
| Age in years | | | |
| 15–19 | 38/318(11.95) | 1.0 | |
| 20–24 | 111/509(21.81) | 1.82[1.30, 2.57]** | 1.33[0.94, 1.89] |
| 25–34 | 199/602(33.06) | 2.77[2.01, 3.81]*** | 1.45[1.00, 2.10]* |
| 35–44 | 117/328(35.67) | 2.99[2.14, 4.16]*** | 1.55[1.05, 2.27]* |
| 45+ | 20/75(26.67) | 2.23[1.38, 3.60]** | 1.07[0.64, 1.79] |
| Marital Status | | | |
| Never Married | 107/755(14.17) | 1.0 | |
| Currently Married | 321/868(36.98) | 2.61[2.15, 3.17]*** | 2.01[1.57, 2.57]*** |
| Previously Married | 57/209(27.27) | 1.92[1.45, 2.55]*** | 1.41[1.02, 1.95]* |
| Hiv Status | | | |
| Negative | 396/1,558(25.42) | 1.0 | 1.0 |
| Positive | 89/274(32.48) | 1.28[1.06, 1.55]* | 0.95[0.78, 1.15] |
| Sexual Partners in the past 12 months | | | |
| One Partner in the past 12 months | 206/913(22.56) | 1.0 | 1.0 |
| More Than One Partner in the past 12 months | 259/827(31.32) | 1.49[1.26, 1.76]*** | 1.36[1.15, 1.61]*** |
| Occupation | | | |
| Trading Center | 71/301(23.59) | 1.0 | 1.0 |
| Agrarian | 242/991(24.42) | 1.04[0.82, 1.30] | 1.08[0.87, 1.34] |
| Fishing | 172/540(31.85) | 1.35[1.06, 1.71]* | 1.10[0.88, 1.39] |
| Education | | | |
| Post-Primary | 119/585(20.34) | 1.0 | |
| Primary | 343/1,184(28.97) | 1.42[1.19, 1.71]*** | 1.29[1.08, 1.53]** |
| No formal education | 23/63(36.51) | 1.79[1.25, 2.58]** | 1.36[0.96, 1.92] |

***p<0.001,

**p<0.01 &

*p<0.05

recommended WHO target of zero [20]. This puts men at risk of complications and potentially HIV acquisition and transmission to female partners [10, 20]. In 2018–2020, we noted significant proportion differences in terms of age; 15–24 years, education [especially post-primary], differences in occupations including the largest decrease in ESR compared to the previous survey rounds. The potential explanation is that, there was generally low participation in visit four, which was potentially caused by the COVID-19 lockdown where people to people movement was strictly prohibited, including those using their private means, hence affecting participation in the survey-round activities.

We observed a relationship between marital status and ESR. Married men were consistently more likely to report ESR than unmarried men consistent with at least one other study [17]. This could be partly due to pressure from wives [24]. This result is also consistent with results

from a qualitative study where men reported fear of losing their wives as a key driver to them resuming sex early [10].

Additionally, an association was observed between age and ESR following VMMC. Men aged 20–24 years and those aged 25–34 years were more likely to resume sex early compared to those aged 15–19 years. This finding is consistent with previous research conducted in Kenya [7, 8]. The explanation for this association could be because more men aged 20–24 years are sexually active than males aged 15–19 years. Strong post-circumcision counseling for older men is critical for controlling HIV incidence and prevalence in these populations [25].

Furthermore, participants who had multiple sexual partners were more likely to resume sex early compared to those who had one sexual partner. This finding is consistent with findings from Kenya. Health education and sensitization strategies that encourage complete wound healing and other safe sexual practices are recommended for men with multiple sexual partners to benefit from the benefits accrued from VMMC.

A limitation to this study is that sexual resumption questions were based on self-reports and is likely to be under-reported due to social desirability. Additionally, we think that VMMC status being only determined by self-report is could have affected these findings. Secondly, we could not tell if the client was HIV positive at the time of VMMC service uptake. Thirdly, we acknowledge recall bias since we included clients who had been circumcised almost 3 years ago. Further qualitative research to understand client and partner attitudes about early sexual resumption would be of value.

## Conclusions

Generally, ESR declined between 2013 and 2020; and was more common among men who are married, had multiple sex partners, and lower levels of education. More strategic interventions are needed to reduce ESR among these men. Strategies like counseling, female partner participation, reminders in form of messages should be encouraged, intensive sensitization, health education, counseling services tailored to reduce prevalence of ESR have been proposed in this study.

## Acknowledgments

We thank Rakai Health Sciences Program (RHSP) for availing the data. We would like to extend our gratitude to our RCCS participants for sparing time and participate in the study.

## Author Contributions

**Conceptualization:** Alex Daama, Fred Nalugoda, Edward Kankaka, Godfrey Kigozi.

**Data curation:** Alex Daama, Fred Nalugoda, Edward Kankaka, Asani Kasango, Betty Nantume, Grace Nalwoga Kigozi, Robert Ssekubugu, Juliana Namutundu, Absalom Ssettuba, Tom Lutalo, Joseph Kagaayi, Gertrude Nakigozi, Ron Gray, Godfrey Kigozi.

**Formal analysis:** Alex Daama, Edward Kankaka, Absalom Ssettuba.

**Funding acquisition:** David Serwadda.

**Investigation:** Joseph Kagaayi, Maria Wawer, David Serwadda, Nelson Sewankambo, Godfrey Kigozi.

**Methodology:** Alex Daama, Fred Nalugoda, Edward Kankaka, Betty Nantume, Grace Nalwoga Kigozi, Robert Ssekubugu, Juliana Namutundu, Absalom Ssettuba, Joseph Kagaayi,

Gertrude Nakigozi, Stella Alamo, Lisa A. Mills, Geoffrey Kabuye, Ron Gray, Maria Wawer, David Serwadda, Nelson Sewankambo, Godfrey Kigozi.

**Resources:** David Serwadda.

**Supervision:** Fred Nalugoda, Asani Kasango, Betty Nantume, Grace Nalwoga Kigozi, Robert Ssekubugu, Juliana Namutundu, Tom Lutalo, Gertrude Nakigozi, Stella Alamo, Geoffrey Kabuye, Ron Gray, Godfrey Kigozi.

**Validation:** Alex Daama, Asani Kasango, Absalom Ssettuba, Joseph Kagaayi, Stella Alamo, Lisa A. Mills, Ron Gray, Maria Wawer, David Serwadda, Nelson Sewankambo.

**Visualization:** Alex Daama, Fred Nalugoda, Asani Kasango, Grace Nalwoga Kigozi, Robert Ssekubugu, Tom Lutalo, Stella Alamo, Lisa A. Mills, Geoffrey Kabuye, Ron Gray, Maria Wawer, David Serwadda, Nelson Sewankambo, Godfrey Kigozi.

**Writing – original draft:** Alex Daama, Godfrey Kigozi.

**Writing – review & editing:** Alex Daama, Fred Nalugoda, Edward Kankaka, Asani Kasango, Betty Nantume, Grace Nalwoga Kigozi, Robert Ssekubugu, Juliana Namutundu, Absalom Ssettuba, Tom Lutalo, Joseph Kagaayi, Gertrude Nakigozi, Stella Alamo, Lisa A. Mills, Geoffrey Kabuye, Ron Gray, Maria Wawer, David Serwadda, Nelson Sewankambo, Godfrey Kigozi.

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
