## [Decision Letter · Decision Letter 0]

13 Feb 2024

PONE-D-23-40397RESUMPTION OF SEXUAL ACTIVITY AFTER VOLUNTARY MEDICAL MALE CIRCUMCISION: DATA FROM A POPULATION BASED COHORT  IN RAKAI, UGANDA, 2013 TO 2020PLOS ONE

Dear Dr. DAAMA,

Thank you for submitting your manuscript to PLOS ONE. After careful consideration, we feel that it has merit but does not fully meet PLOS ONE’s publication criteria as it currently stands. Therefore, we invite you to submit a revised version of the manuscript that addresses the points raised during the review process.

We look forward to receiving your revised manuscript.

Kind regards,

Hamufare Dumisani Dumisani Mugauri, Ph.D. Public Health

Academic Editor

PLOS ONE

Journal Requirements:

2. In the online submission form, you indicated that [All data files will be available upon request.]. 

Reviewers' comments:

Reviewer's Responses to Questions

**Comments to the Author**

1. Is the manuscript technically sound, and do the data support the conclusions?

Reviewer #1: Yes

Reviewer #2: Yes

2. Has the statistical analysis been performed appropriately and rigorously? 

Reviewer #1: Yes

Reviewer #2: Yes

3. Have the authors made all data underlying the findings in their manuscript fully available?

Reviewer #1: No

Reviewer #2: No

4. Is the manuscript presented in an intelligible fashion and written in standard English?

Reviewer #1: Yes

Reviewer #2: No

5. Review Comments to the Author

Reviewer #1: Title: The title is ok. However perhaps word "early resumption" rather than just resumption of sexual activity is what is being studied so this should be inserted. The study method needs to be added - a cross sectionals survey

Abstract: The Abstract strikes me as been too lengthy and lacking in flow, there are too many brackets which interrupt the flow of the text, making the abstract difficult to read and assimilate easily. These should be minimised and only the essential information need be provided.

Introduction: This is mainly providing the justification for the study, without stating the background of the study population and the study objective. This should included some description of RCCS which is instead in the study methods. The study objective should be provided at the end of the introduction, this is missing.

Methods: The is some disparity between the WHO recommendation of 6 weeks to resume sexual activity and the MOH of Uganda's recommendation of 42 days. Some comments and justification should be provided on this. It is a little unclear how if the study is presumed to have been done in 2024 or 2023 how recent circumcision (<3 years) would have included those who had VMMC in cohorts (2013-2015,2016 to 2018 etc). Unless these surveys were summated, this should be made clear. It is not also clear if these participants were surveyed at the study time or the information was just pooled from the previous surveys in data base, because there is an issue of memory or recall Biase then in these participants. The authors need to make this clearer.

Results: The descriptive data which provides general information of the study population is missing. The results go straight into the analytic data. The results are presented in an unyielding manner with long tables with statistical numbers which give the impression of raw data. The results should be presented better and some descriptive data should be provided.

Discussion: This discussion is good. However the explanation that the decline in ESR is based on improvement in counselling services is not supported by this study itself. The author ought to be able to show that this is the case in this study based on their own data. Some discussion on early resumption sex with condom use may be useful

Conclusions

The conclusion is fair and accurate. However it adds little to what is already published, given the large cohort and the period studied the conclusion can be strengthen further.

Reviewer #2: Reviewer comments

1. The title should be “Population-based Cohort…”.

2. Line 7-8: A comma is missing: “Data from the Rakai Community Cohort Study, a cross-sectional study, were analyzed”. Also you refer to men aged 15-49 years, but men are aged ≥18 years. I’d recommend you refer to these clients as “males”.

3. Line 9-10: You should delete the parentheses around the years of these studies.

4. Line 13: For clarity with the 4 surveys, I would refer to this as “…participated in this analysis.”

5. Line 14: Previously you refer to 4 successive surveys, but here you refer to 3 surveys. Maybe say: “Across the first three surveys…” if that is what you mean here.

6. Line 20: The word “years” is missing after “45,” and there is a word missing, as the rest of this sentence does not make sense.

7. Line 21: Is the Adjusted Prevalence Ratio here 0? Also please check the CIs in lines 21-22, as something is wrong here.

8. Line 32: A comma is missing before “which”.

9. Line 33: The word “for” is missing after “allow”.

10. Line 42: These references are numbered out of order.

11. Line 47: This is an 8-year period.

12. General comment about the introduction section: There are several recent publications about sexual resumption following VMMC that have been excluded. I would recommend adding these. I also have a lot of remaining questions after reading the introduction like how many males have been circumcised for HIV prevention in Uganda? What is the specific guidance that circumcised males are given? Are there any particular challenges to Uganda that are related to wound healing or resumption of sexual activity?

13. Line 51: A comma is missing before “which”.

14. Line 54-55 says that the men self-reported their circumcision status, but in line 62 refers to only medically circumcised men. It’s unclear whether these males were identified through self-report or through medical records confirming their VMMC.

15. Line 85: Is the full name of this IRB the “Western IRB”?

16. Line 95: The word “years” is missing after “35-44”.

17. Line 99: “Fishermen” is one word.

18. Line 110: I’m not sure if this is a paragraph or a title.

19. Line 110-119: This doesn’t read as a cohesive paragraph in a Results Section.

20. Line 166: You should say “two-fold,” and this is an 8-year period.

21. Line 179: Reference #19 is missing.

22. Line 181-187: I would assume that more men aged 20-24 years are sexually active than males aged 15-19 years. Are there any data from Uganda that you could cite to support this?

23. General comment about the introduction and discussion section: More HIV-positive males that I would have expected were reported to have been circumcised and resumed sexual activity early. This is noteworthy as VMMC is usually promoted to males who are HIV-negative, though negative status is not required in order to be circumcised. The process for HTS before VMMC should be added to the introduction, and the fact that a high proportion of HIV-positive men were included in this analysis should be mentioned in the discussion section.

24. Line 193: If you think that VMMC status being only determined by self-report is a limitation, then that should be included in this paragraph.

25. Line 236: Reference #5 is incomplete.

26. This manuscript needs careful reviewing and proofreading throughout. Several sections lack specificity and detail, so more information is needed in the introduction and methods sections, especially. The results also needs careful attention.

6. PLOS authors have the option to publish the peer review history of their article (what does this mean?). If published, this will include your full peer review and any attached files.

Reviewer #1: **Yes: **kasonde Bowa

Reviewer #2: No

---

## [Author Response · Author response to Decision Letter 0]

27 Jul 2024

Point by point 

S/N Comments Solution 

SPECIFIC COMMENTS: 

1 Reviewer #1: Title: The title is ok. However perhaps word "early resumption" rather than just resumption of sexual activity is what is being studied so this should be inserted. The study method needs to be added - a cross sectionals survey The title has been re-written having both early and the methods

2 Abstract: The Abstract strikes me as been too lengthy and lacking in flow, there are too many brackets which interrupt the flow of the text, making the abstract difficult to read and assimilate easily. These should be minimized and only the essential information need be provided. The abstract has been refined and some brackets have been removed to ease the flow and understanding 

3 Introduction: This is mainly providing the justification for the study, without stating the background of the study population and the study objective. This should included some description of RCCS which is instead in the study methods. The study objective should be provided at the end of the introduction, this is missing. The RCCS description has been added and the overall objective included. These portions were removed from the methods section.

4 Methods: The is some disparity between the WHO recommendation of 6 weeks to resume sexual activity and the MOH of Uganda's recommendation of 42 days. Some comments and justification should be provided on this. It is a little unclear how if the study is presumed to have been done in 2024 or 2023 how recent circumcision (<3 years) would have included those who had VMMC in cohorts (2013-2015, 2016 to 2018 etc). Unless these surveys were summated, this should be made clear. It is not also clear if these participants were surveyed at the study time or the information was just pooled from the previous surveys in data base, because there is an issue of memory or recall Biase then in these participants. The authors need to make this clearer. Thank you so much for this important comment. Yes, there is a clear difference between WHO and Ugandan MOH definitions 6 weeks vs 42 days. However, for this particular study, our outcome was defined based on the Ugandan MOH definition of 42 days.

The recent circumcision (<3 years) was based on the particular year of the survey e.g survey one happened between 2013 and 2015 therefore the recent circumcision was <3 years on the day of interview or data collection for each survey visit. The most recent survey happened between 2018 and 2020. We have included this in our definition We understand and acknowledge this limitation (recall bias) and we have included this under study limitation.

5 Results: The descriptive data which provides general information of the study population is missing. The results go straight into the analytic data. The results are presented in an unyielding manner with long tables with statistical numbers which give the impression of raw data. The results should be presented better and some descriptive data should be provided. Thank you so much for this observation. We have included the descriptive data for the study population.

6 Discussion: This discussion is good. However the explanation that the decline in ESR is based on improvement in counselling services is not supported by this study itself. The author ought to be able to show that this is the case in this study based on their own data. Some discussion on early resumption sex with condom use may be useful Thank you for this comment. We have adjusted this to fit our own data.

7 Conclusions

The conclusion is fair and accurate. However it adds little to what is already published, given the large cohort and the period studied the conclusion can be strengthen further. Thanks, we have improved on the conclusion 

Reviewer 1 

1 The title should be “Population-based Cohort…” This has been changed to fit the suggestion

2 Line 7-8: A comma is missing: “Data from the Rakai Community Cohort Study, a cross-sectional study, were analyzed”. Also you refer to men aged 15-49 years, but men are aged ≥18 years. I’d recommend you refer to these clients as “males”. Thanks a lot. This has been addressed 

3 Line 9-10: You should delete the parent heses around the years of these studies. Thanks, this has been removed

5 Line 13: For clarity with the 4 surveys, I would refer to this as “…participated in this analysis.” This has been corrected

6 Line 14: Previously you refer to 4 successive surveys, but here you refer to 3 surveys. Maybe say: “Across the first three surveys…” if that is what you mean here. Thanks. This has been addressed 

7 Line 20: The word “years” is missing after “45,” and there is a word missing, as the rest of this sentence does not make sense. Thank you so much. This has been corrected.

8 Line 21: Is the Adjusted Prevalence Ratio here 0? Also please check the CIs in lines 21-22, as something is wrong here. Yes, the Prevalence Ratios are 0 and the CIs are also 0.0000018 and 0.0000026

9 Line 32: A comma is missing before “which”. Thank you , a comma has been added

10 Line 33: The word “for” is missing after “allow”. Thank you, the word for has been added

11 Line 42: These references are numbered out of order. 

12 Line 47: This is an 8-year period. This was corrected as per tracked changes copy

13 General comment about the introduction section: There are several recent publications about sexual resumption following VMMC that have been excluded. I would recommend adding these. I also have a lot of remaining questions after reading the introduction like how many males have been circumcised for HIV prevention in Uganda? What is the specific guidance that circumcised males are given? Are there any particular challenges to Uganda that are related to wound healing or resumption of sexual activity? The introduction section has been refined 

Circumcised men in Uganda are advised to resume sex after 42 days or 6 weeks (each week having 7 days) as per ministry of health guidelines.

14 Line 51: A comma is missing before “which”. Corrected

15 Line 54-55 says that the men self-reported their circumcision status, but in line 62 refers to only medically circumcised men. It’s unclear whether these males were identified through self-report or through medical records confirming their VMMC. All the information about circumcision was self-report by the participants.

We have corrected this in the main document. 

16 Line 85: Is the full name of this IRB the “Western IRB”? Johns Hopkins University School of Medicine

17 Line 95: The word “years” is missing after “35-44”. The word years has been added 

18 Line 99: “Fishermen” is one word. Thanks, this has been addressed

19 Line 110: I’m not sure if this is a paragraph or a title. This is a title and has been revised 

20 Line 110-119: This doesn’t read as a cohesive paragraph in a Results Section. This has been improved 

21 Line 166: You should say “two-fold,” and this is an 8-year period. This has been addressed 

22 Line 179: Reference #19 is missing. Thanks. The reference 

23 Line 181-187: I would assume that more men aged 20-24 years are sexually active than males aged 15-19 years. Are there any data from Uganda that you could cite to support this? Thanks, actually your assumption is correct and we have no data in Uganda that disagrees with that assumption.

This has been corrected 

24 General comment about the introduction and discussion section: More HIV-positive males that I would have expected were reported to have been circumcised and resumed sexual activity early. This is noteworthy as VMMC is usually promoted to males who are HIV-negative, though negative status is not required in order to be circumcised. The process for HTS before VMMC should be added to the introduction, and the fact that a high proportion of HIV-positive men were included in this analysis should be mentioned in the discussion section The information has been added in the introduction

25 Line 193: If you think that VMMC status being only determined by self-report is a limitation, then that should be included in this paragraph. Thanks, this has been addressed 

26 Line 236: Reference #5 is incomplete. This has been rectified. Thanks

27 This manuscript needs careful reviewing and proofreading throughout. Several sections lack specificity and detail, so more information is needed in the introduction and methods sections, especially. The results also needs careful attention. Thanks

---

## [Decision Letter · Decision Letter 1]

27 Aug 2024

PONE-D-23-40397R1POPULATION-BASED COHORT DATA USED TO ASSESS TRENDS IN EARLY RESUMPTION OF SEXUAL ACTIVITY AFTER VOLUNTARY MEDICAL MALE CIRCUMCISION IN RAKAI, UGANDA.PLOS ONE

Dear Dr. DAAMA,

Thank you for submitting your manuscript to PLOS ONE. After careful consideration, we feel that it has merit but does not fully meet PLOS ONE’s publication criteria as it currently stands. Therefore, we invite you to submit a revised version of the manuscript that addresses the points raised during the review process.

We look forward to receiving your revised manuscript.

Kind regards,

Hamufare Dumisani Dumisani Mugauri, Ph.D. Public Health

Academic Editor

PLOS ONE

Reviewers' comments:

Reviewer's Responses to Questions

**Comments to the Author**

1. If the authors have adequately addressed your comments raised in a previous round of review and you feel that this manuscript is now acceptable for publication, you may indicate that here to bypass the “Comments to the Author” section, enter your conflict of interest statement in the “Confidential to Editor” section, and submit your "Accept" recommendation.

Reviewer #1: All comments have been addressed

Reviewer #2: (No Response)

2. Is the manuscript technically sound, and do the data support the conclusions?

Reviewer #1: Yes

Reviewer #2: Partly

3. Has the statistical analysis been performed appropriately and rigorously? 

Reviewer #1: Yes

Reviewer #2: Yes

4. Have the authors made all data underlying the findings in their manuscript fully available?

Reviewer #1: Yes

Reviewer #2: Yes

5. Is the manuscript presented in an intelligible fashion and written in standard English?

Reviewer #1: Yes

Reviewer #2: Yes

6. Review Comments to the Author

Reviewer #1: The authors have adequately address all the key comments in the previous reviews in the areas of the abstract, the introduction, the methods, the results, the discussion and conclusions drawn

Reviewer #2: 1. Typos in the Methods and Results sections of the Abstract

2. In line 21, it’s awkward that you refer to men with “no sexual relationship” reporting resuming sexual activities. I’d recommend you clarify the language to make it clear that you’re referring to men who were not married or otherwise in relationships.

3. In line 34, your references are incorrect. You are missing the Gray manuscript from 2007, the Auvert manuscript from 2004, and you should include the meta-analysis by Weiss from 2000. The current reference #2 looks to be incorrect as well.

4. In line 34, you must add that VMMC reduces the risk of heterosexual HIV acquisition by 50-60% in men.

5. I’m not familiar with the recommendations by UNICEF, as mentioned in line 35. This should be cited or deleted.

6. It’s unclear if a new paragraph is supposed to start in line 56.

7. The introduction section is lacking some important information to orient the reader to VMMC and HIV prevention efforts in Uganda. The authors should include approximately how many males and the proportion of males that have been circumcised for HIV prevention in Uganda, what is the specific post-operative guidance that circumcised males are given regarding sexual activity post-VMMC (e.g., no vaginal sex, no sexual activity at all, no masturbation, etc.) Are there any particular challenges to Uganda that are related to wound healing or resumption of sexual activity that make adherence to the guidance particularly difficult? Note that this recommendation was included in the first review of this manuscript as comment #13.

8. I assume that boys aged <18 years provided assent, not written consent, as noted in line 80. Please clarify the language here if the current language is not right. Same comment in line 100.

9. See the original comment #2. Many study participants are aged <18 years, so referring to participants as “men” is incorrect. This issue has not been corrected throughout the manuscript.

10. The paragraph from rows 104-120 contains many errors that makes it difficult to read and understand. The use of the terms “consistent” and “inconsistent” is awkward here. Perhaps readability would be improved if the authors used “similar” and “dissimilar”.

11. The sentence in line 132 does not make sense.

12. What is R19 in line 134?

13. Figure 1 (lines 175-176) has two titles.

14. In Table 1, it appears as though survey round 4 (2018-2020) was significantly different than previous rounds (i.e., younger cohort, more post-primary education, differences in occupations, etc.). Also the largest decrease in ESR occurs in survey round 4. Is it possible that these cross-sectional cohorts include younger men that are different than prior surveys? The possible reasons for this should be discussed in the discussion section.

15. What is the target that is mentioned in line 242? This should be included here.

16. The list of references look inconsistent with each other and with the journal’s instructions.

17. In general, this manuscript still has several issues. There are many typos and awkward word choices that affect its readability. Careful review and copyediting is needed before resubmission.

7. PLOS authors have the option to publish the peer review history of their article (what does this mean?). If published, this will include your full peer review and any attached files.

Reviewer #1: **Yes: **Kasonde Bowa

Reviewer #2: No

---

## [Author Response · Author response to Decision Letter 1]

26 Oct 2024

S/N Comments Solution 

SPECIFIC COMMENTS: 

1 Reviewer #1: The authors have adequately address all the key comments in the previous reviews in the areas of the abstract, the introduction, the methods, the results, the discussion and conclusions drawn 

Thanks 

2 Reviewer #2: 1. Typos in the Methods and Results sections of the Abstract We have corrected typos in the methods and results sections

3 In line 21, it’s awkward that you refer to men with “no sexual relationship” reporting resuming sexual activities. I’d recommend you clarify the language to make it clear that you’re referring to men who were not married or otherwise in relationships. We corrected this, the confusing statements removed.

4 In line 34, your references are incorrect. You are missing the Gray manuscript from 2007, the Auvert manuscript from 2004, and you should include the meta-analysis by Weiss from 2000. The current reference #2 looks to be incorrect as well. We have corrected this and added these new references

5 In line 34, you must add that VMMC reduces the risk of heterosexual HIV acquisition by 50-60% in men. This has been added

6 I’m not familiar with the recommendations by UNICEF, as mentioned in line 35. This should be cited or deleted. This was sorted 

7 It’s unclear if a new paragraph is supposed to start in line 56. Yes this was a new paragraph

8 The introduction section is lacking some important information to orient the reader to VMMC and HIV prevention efforts in Uganda. The authors should include approximately how many males and the proportion of males that have been circumcised for HIV prevention in Uganda, what is the specific post-operative guidance that circumcised males are given regarding sexual activity post-VMMC (e.g., no vaginal sex, no sexual activity at all, no masturbation, etc.) Are there any particular challenges to Uganda that are related to wound healing or resumption of sexual activity that make adherence to the guidance particularly difficult? Note that this recommendation was included in the first review of this manuscript as comment #13. This information has been included in the introduction 

9 I assume that boys aged <18 years provided assent, not written consent, as noted in line 80. Please clarify the language here if the current language is not right. Same comment in line 100. This has been clarified 

10 See the original comment #2. Many study participants are aged <18 years, so referring to participants as “men” is incorrect. This issue has not been corrected throughout the manuscript. This has been addressed 

11 The paragraph from rows 104-120 contains many errors that makes it difficult to read and understand. The use of the terms “consistent” and “inconsistent” is awkward here. Perhaps readability would be improved if the authors used “similar” and “dissimilar”. Addressed 

12 The sentence in line 132 does not make sense. Addressed 

13 What is R19 in line 134? Removed 

14 Figure 1 (lines 175-176) has two titles. Addressed 

15 In Table 1, it appears as though survey round 4 (2018-2020) was significantly different than previous rounds (i.e., younger cohort, more post-primary education, differences in occupations, etc.). Also the largest decrease in ESR occurs in survey round 4. Is it possible that these cross-sectional cohorts include younger men that are different than prior surveys? The possible reasons for this should be discussed in the discussion section. We have included a potential explanation in discussion section

16 What is the target that is mentioned in line 242? This should be included here. Basically, WHO recommends zero persons resuming sex before the recommended period

17 The list of references look inconsistent with each other and with the journal’s instructions. Corrected 

18 In general, this manuscript still has several issues. There are many typos and awkward word choices that affect its readability. Careful review and copyediting is needed before resubmission. Addressed

---

## [Editor Report · Decision Letter 2]

5 Nov 2024

POPULATION-BASED COHORT DATA USED TO ASSESS TRENDS IN EARLY RESUMPTION OF SEXUAL ACTIVITY AFTER VOLUNTARY MEDICAL MALE CIRCUMCISION IN RAKAI, UGANDA.

PONE-D-23-40397R2

Dear Dr. Daama,

We’re pleased to inform you that your manuscript has been judged scientifically suitable for publication and will be formally accepted for publication once it meets all outstanding technical requirements.

Kind regards,

Hamufare Dumisani Mugauri, Ph.D. Medicine and Health Sciences

Academic Editor

PLOS ONE
---

## [Editor Report · Acceptance letter]

12 Nov 2024

PONE-D-23-40397R2 

PLOS ONE

Dear Dr. DAAMA, 

I'm pleased to inform you that your manuscript has been deemed suitable for publication in PLOS ONE. Congratulations! Your manuscript is now being handed over to our production team.

Kind regards, 

on behalf of

Mr Hamufare Dumisani Mugauri 

Academic Editor

PLOS ONE